# HESS Opinions

## Unsaturated infiltration: the need for a reconsideration of historical misconceptions

Peter Germann[†]

[†] *Deceased, 6th December 2020*

*Correspondence to:* Keith Beven (k.beven@lancaster.ac.uk)

**Abstract** Briggs (1897) deduced capillary flow from deviation of the equilibrium between capillarity and gravity. Richards (1931) raised capillary flow to an unproven soil hydrological dogma. Attempts to correct the

dogma led to non-equilibrium flow, macropore flow, and preferential flow during infiltration. Viscous film flow is proposed as an alternative approach to capillarity-driven flow during unsaturated infiltration.

**Preface:  Peter F. Germann, in memoriam**

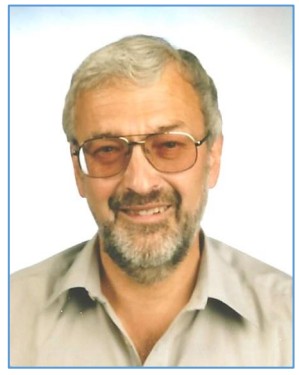


The paper that follows was submitted to HESS by Peter Germann in September 2020.   Unfortunately, before the processing of the paper could be completed, Peter passed away rather suddenly on December 6th 2020 in Bern, Switzerland after two major strokes.   Peter was well known to many soil hydrologists for his work on preferential

flows; a career of research work that was summarised in his 2014 book on the subject published by the University of Bern (Germann, 2013).

Peter and I had been good friends since we first met and started doing field work together at the Institute of Hydrology at Wallingford in 1979 where he spent a year as a post-doc and we found we had a common interest in

preferential flows.   Beven and Germann (2013) includes some information and a photograph of the tracing experiment we did at Grendon Underwood in 1979.   This was instructive in that it was done at the end of the winter on well-wetted soil when all the cracks in the clay should have been closed.   In fact, it took the Rhodamine dye we added at the surface some 60 seconds to reach a mole drain at a depth of 45 cm, and destructive sampling

after the trace showed that the cracks were still transmitting water (and that the grass roots were mostly on the surface of the peds).

Peter was born in St. Gallen Switzerland in 1944. He grew up in Bischofszell and later completed his schooling in St. Gallen. In 1963-1969 he studied for a Degree in Forestry at ETH Zurich, and then stayed with Professor Richard to carry out research for a PhD at the Eidgenössische Forschungsanstalt für Wald, Schnee und Landschaft (WSL) 1969-1976. He continued his work in the Laboratories of Hydraulics, Glaciology and Hydrology (VAW) during the period 1976-1980. His PhD work was a study of the water relations on a forested slope in the Rietholzbach catchment based on maintaining a network of 35 nests of tensiometers at 10 different depths down to 3m, set out on a triangular grid amongst the trees. At this time these were still manual tensiometers coupled to mercury manometers that were read every 2-3 days for 3 years. One of the features that this remarkable data set revealed was that during infiltration wetting could occur at depth in some cases, apparently by-passing the tensiometers above. Another was the large heterogeneity in responses between sites and between wetting events.

Peter returned to Switzerland from Wallingford and then in 1980 he took up a post as Assistant Professor in the Department of Environmental Sciences at the University of Virginia (UVA) in Charlottesville where we overlapped again for 2 years. Peter stayed at UVA until 1986. He then moved as an Associate Professor to the Department of Soils and Crops at Rutgers University. In 1989 he was offered a Professorship at the Institute of Geography, University of Bern back in Switzerland where he stayed until he retired in 2009. He held an Emeritus position at Bern until 2015.

For the major part of his research career, Peter was a strong advocate for a reconsideration of the physics of water flow through soils and, in particular, for the limitations of the Darcy-Buckingham-Richards flow theory that is based on an assumption of the equilibration of capillary potentials in some (not clearly defined) "representative elementary volume" of soil pores. The paper that follows is a continuation of that theme.

Our common interest resulted in the highly cited review paper in Water Resources Research on Macropores and Water Flow in Soils (Beven and Germann , 1982), and a sequence of 3 papers in J. Soil Science (Germann and Beven, 1981a,b; Germann and Beven, 1981). including a modelling approach based on kinematic wave theory that received very critical reviews. Some soil physicists at that time appeared to believe that the physics of soil physics had been solved (and that is still a common belief and still common in the teaching of soil hydrology). In 1985,

we both attended an International Workshop on Water and Solute Movement in Heavy Clay Soils, held in Wageningen, where we presented a paper on flow in distributions of macropores based on the kinematic wave approach.  In the bar one evening we had a long discussion over a beer with John Philip who argued that it would not be possible that film flow in a macropore, which would have a curved meniscus and therefore a potential at less than atmospheric pressure, could result in an outflow at the base of an undisturbed column since a positive

pressure would be required.  I have never been able to decide whether John was really serious in his argument, or whether he was just teasing these two earnest young men committed to the importance of preferential flows.

Peter later developed the kinematic wave approach into a theory of viscosity (rather than capillarity) dominated film flows subject to Stokes' law during infiltration.  This is still somewhat contentious (see the comment on the

paper by John Nimmo at https://hess.copernicus.org/preprints/hess-2020-499/), and Peter rather glossed over issues such as the change of parameters that might be expected as smaller pores and pore necks involved in the film flow became filled and that require some effective parameter values.  As he demonstrates in his book, he felt that the approach was supported by both the field and laboratory evidence.  Peter was a careful experimentalist and made use of a variety of time domain reflectometry, sonic and tracer experiments to study preferential flows.

His thoughtful and good-humoured approach to collaborations, discussions and presentations will be missed by many in soil science.  A more complete biography and account of Peter's contributions can be found at http://www.history-of-hydrology.net/mediawiki/index.php?title=Germann,_Peter

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

## 1 Introduction

Why does water know / where to flow? / is the strongest force / dictating the course? / Is the weakest resistance / controlling the distance? / Or is soil hydrology nothing but mental strength / fiddling with mass, time, and length? (Germann, 2014)

## 2 How capillary flow got its position in unsaturated porous media flow

The success of terrestrial plants relies, among other phenomena, on the simultaneous supply of oxygen and water to the root tips in the range of < 5 to > 50 μm. A mandatory prerequisite is the inferior water pressure relative to the air pressure around the tips. That again is the result of the water's surface tension against the air and its affinity towards the solid particles, resulting in the capillary potential $\psi$. As a physicist with the US-Bureau of Soils, Lyman Briggs (1897) formalized the relationships mathematically and physically. He made use of data from a sand column experiment, reported in Franklin King (1892), 42 inches high, that was completely water saturated and then left to drain for 40 days. He qualitatively expressed an equilibrium between the gravitational and capillary forces, represented by height and pressure drop across the water meniscus, deducing that moisture content will decrease with height in the column.

Briggs also provides a qualitative description for flow when there is any deviation from this static equilibrium in terms of a capillary gradient, and further postulated a variable permeability concept in partially water saturated soils. He certainly introduced these concepts to Edgar Buckingham who joined the Bureau of Soils to work under Briggs in 1902. Buckingham further developed these concepts and his reports on the movement of soil moisture is frequently viewed as the origin of modern concepts of capillary flow (Buckingham, 1904, 1907): He extended a first-order gradient flux law to unsaturated soils as analogous to thermal and electrical fluxes (though does not explicitly mention Darcy's law), and he proposed hydraulic conductivity, diffusivity and the water capacity as quantitative functions of soil moisture. However, "... *Buckingham did not manage to formulate a clear-cut physical-mathematical flow theory that quickly inspired other soil physicists*" (Raats and Knight, 2018, p. 11).

This seminal work at the Bureau of Soils was set back by the Chief of the Bureau, Milton Whitney (who was Chief from 1860 to his retirement in 1927). Whitney dismissed King in 1904 because of his belief that soil nutrient status were less important to crop growth than the soil physical properties favoured by Whitney. He also encouraged Briggs to leave in 1905 because his training as a physicist was resulting in the application of "*rigid mathematical demonstration*" to soil physics that Whitney felt could not be justified (see Landa and Nimmo, 2003). After the departure of Briggs, Buckingham worked under Frank Kenneth Cameron who was not totally convinced by Buckingham's developments, suggesting that there might be inconsistencies in Buckingham's approach to water and vapour transport. Buckingham's response was not totally convincing, but Cameron decided not to hold back the publication as "*you seem to be convinced that it is all sound and safe*" (quoted in Nimmo and Landa, 2005). Buckingham's 1907 report was delayed as a consequence until after he had also left the Bureau of Soils

to join the National Bureau of Standards in August 1906. He never returned to the study of soil physics. He became much better known as the author of the Π theorem for dimensional analysis (Buckingham, 1914) which was only much later applied to soil moisture characteristics (e.g. Miller and Miller, 1956).

It was more than two decades before there was further significant progress with Lorenzo K. Richards' (1931) paper on the "Capillary Conduction of Liquids Through Porous Mediums", where he presented the three-dimensional form of the second-order convection-diffusion equation, that bears his name. Note that, despite the comment of Raats and Knight (2018) quoted above, all the elements of the Richards equation were already present in the report of Buckingham (1907), but he did not there combine them into a single flow equation. In fact, Richardson (1922)
had already done so before Richards but outside the context of the soil physics community (see Raats and Knight, 2018). Richards did, however, also present an experimental procedure for the determination of the parameters of his equation, including a demonstration of hysteresis in the soil water characteristics. This combination of theory and experiment based on that experiment made the approach attractive for practical applications.

However, Richards was also instrumental in introducing some misconceptions into soil physics. He authoritatively stated "*If there is a steady flow of liquid through a porous medium which is only partially saturated, then the larger pore space containing air and the effective cross-sectional area of the water conducting region is reduced. If these air spaces could in some way be filled with solid, the condition of the flow would be unchanged and the proportionality between the flow and the water-moving force would still hold because Darcey's law is*
*independent of the size of particles or the state of packing*" (Richards, 1931, p.323). This statement reveals the fundamental, but scientifically quite wrong, exclusivity he put on capillarity-driven flow during infiltration. Since then it has been raised to a soil hydrological dogma that still confronts us today. In the theory of Richards, increasing the flow rate in a partially saturated porous medium mandatorily requires the increase of the hydraulic gradient by locally either increasing or decreasing capillary potential that, in turn, has to equilibrate in either case
with the water content. In Richards' context this is only feasible during infiltration when flow remains sequential i.e., finer pores have to fill before coarser pores are allowed to (Germann, 2018). Indeed, it has been suggested that the Richards' experiment, that relied on the application of air pressure to ensure that the larger pores were empty at each potential and flow rate, was simply the wrong experiment for flows under more natural conditions (Beven, 2014, 2018).

### 3 Alternative approach to infiltration and drainage
A reconsideration of the ubiquity of Richards' (1931) capillary approach to infiltration and drainage demands a concept that replaces this assumption of dominance by capillarity. In early attempts, Germann and Beven (1981), Beven and Germann (1981) and Germann (1985) successfully approached transient infiltration/drainage with
kinematic wave theory that can be applied when gravity is the dominant driving force rather than the gradient of capillary potential, which will be the strongest driver under drier conditions. More recently, Germann and al Hagrey (2008) summarize the features of gravity-dominated viscous film flow[1] during transient infiltration and

---

[1] John Nimmo in his comments on this paper (see https://hess.copernicus.org/preprints/hess-2020-499/) points out that there is problem in Peter Germann's use of "viscous flow" to describe his film flow concepts since viscosity is intrinsic to all types of flow, including flows that conform to the Buckingham-Richards theory where, viscosity is intrinsic to the definition of hydraulic conductivity albeit that it often does not appear explicitly in expositions

drainage in the Kiel sand tank. (i) constant velocity of the wetting front; (ii) collapse to atmospheric pressure of capillarity behind the wetting front; (iii) infiltration and drainage follow the same simple rules of viscous film flow. Moreover, no pore classification or size distribution or consideration of heterogeneity of soil properties is required to the application of viscous film flow, since film thickness, flow velocity and celerity during both wetting and drying can be related to flux rates.

**4 What happens at the soil surface during infiltration?**

Infiltration expresses how water arriving from above the ground, usually in the shape of drops, transitions to water flow in the soil below the ground surface. Drops have internal pressures higher than atmospheric. According to Laplace-Young theory, pressures against the atmosphere in drops increase from about 15 to 750 Pa as their diameters reduce from 5 to 0.1 mm. When drops hit the ground surface, depending on their size and Bond number, they will often burst. The remnants then join and form local films at atmospheric pressure. There are two ways for water to continue. Either, water has to follow the strongest gradient, as in capillary flow. Richards' (1931) forceful conjecture of the dominance of capillarity during infiltration in unsaturated porous media works well under steady-state conditions. however, increasing the flow rate causes the well-known phenomena of non-equilibrium flow, macropore flow, and preferential flow. Water then follows the way of least resistance, as a viscous film flow. This type of flow occurs under atmospheric pressure, regardless of the thickness and path widths of the films, as Flammer et al. (2000) have demonstrated with acoustic velocity measurements across soil columns. Germann (2018) also demonstrates how acoustic velocity experiments suggest that preferential flows can proceed faster than capillary gradient driven fluxes in unsaturated soils.

**5 Where are we now?**

On the one hand, the Richards' (1931) capillary gradient flow (and implied local equilibration of potentials) continues to be taught and used in models of soil hydrology in a rather dogmatic way. There have been modifications suggested to allow for dual permeability functions near saturation but these are not really convincing (Beven and Germann, 2013; Beven, 2018). On the other hand, there are a plethora of review articles that discuss the hydro-mechanical inconsistencies with the Richards equation and observations of preferential, non-equilibrium, and macropore flows in both unsaturated and saturated soils (see again Beven, 2018 and references therein). This conflict remains to be resolved, but is clearly important for the understanding of both flow and transport. It is, however, the case that there are few ongoing research efforts, either theoretical and experimental, on developing coherent descriptions of preferential flow in soil physics. The author's application of viscous film flow concepts to infiltration in unsaturated soils has been one such effort that it is hoped will be taken on and further developed by others (see, for example, the recent papers of Germann, 2017, 2018a,b; Germann and Prasuhn, 2018; Germann and Karlen, 2016).

       In conclusion: Briggs (1897) convenient and, within his concept correct definition of capillary flow, led via the work of Buckingham to Richards' (1931) dogma of the unproven dominance of capillary gradient driven fluxes during infiltration that is still widely used today. However, as it turns out, while capillary flow relies on

---

of the Richards equation. Viscosity does appear explicitly in the Stokes relations for gravity-dominated film flows. In editing the paper KB has therefore referred to viscous film flow to keep continuity with Peter's usage.

the strongest force exerted on water in an unsaturated soil, the weaker force of viscosity can dominate during infiltration as a result of the formation of film flows at and near the soil surface..

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
