# Peer review of "HESS Opinions"

_Hydrology and Earth System Sciences, 2020_

## Referee Comment (RC1) · John R. Nimmo (Referee) · 19 Jan 2021

The central point of this stimulating commentary, that capillarity does not always dominate unsaturated flow, is crucial to the understanding and prediction of water flow in the unsaturated zone. The lack of universal recognition that gravity often dominates, to the point of overwhelming the potentially stronger force of capillarity, impedes progress in this field. The commentary also highlights three classic papers in the scientific literature and stimulates interest in the insights and perspective to be gained by reading these works. In this review I explain some disagreements with various details and interpretations of the historical and physical topics presented, and with use of certain terminology. Factual errors should be corrected in revising the manuscript, and my comments below should be addressed with appropriate changes or additional discus-

sion.

Describing the work of Briggs (1897), statements in this paper go somewhat too far. Briggs does explain in this report the roles of surface tension and viscosity, and the balance of gravitational and capillary forces. In terms of soils, however, he does so only in a qualitative and comparative way. He briefly mentions permeability, and the importance of viscosity in determining flow rate, but I do not see that he directly mentions hydraulic conductivity of unsaturated soil or a retention curve $\psi(\theta)$. The sand column experiment that Germann mentions is one that Briggs credits to King (1892), noting how much drier the column is at the top than at the bottom, but not directly connecting it to a retention curve.

The $\psi(\theta)$ relation, and the correspondence between $\psi$ and height within an equilibrated soil column, are presented quantitatively and in detail by Buckingham (1907), who had worked with Briggs and built on his insights. My understanding of Buckingham's (1907) paper differs in some ways from comments in Germann's commentary and other publications. I would state that Buckingham did indeed develop a quantitative physical theory of unsaturated flow; it lacked only the single-formula mathematical statement of the physical theory that was later supplied by Richardson (1922) and Richards (1931). On page 51 of Buckingham's report, having already presented the Darcy-Buckingham law of unsaturated flow, the dependence of hydraulic conductivity on water content, and the functional relationship between water content and matric potential, he could have simply combined them mathematically to get a three-variable quantification of the transient state and flow of water in unsaturated soil in a single formula.

Germann is basically correct in his assessment of Richards' outsize role in enshrining the concept of capillarity as exclusively dominant in unsaturated flow. Richards' (1931) exposition of how capillarity combines with the conduction-related elements of viscosity, geometry, and potential gradients to yield a single equation for transient flow in a porous medium is clear and well-argued. In the light of Briggs' and Buckingham's earlier work, however, these theoretical contributions of Richards are more modest

than what he is commonly credited with. (Though not an issue in Germann's commentary, there is an irony in that the experimental components of Richards' paper are a widely underemphasized achievement.) More to the point of Germann's commentary, Richards' explication of capillary flow and the prevailing interpretations of it over the last ninety years carry an implication that it is the only type of unsaturated flow that needs to be considered. Germann's rejection of that is clear. The commentary would be strengthened by clarifying what the appropriate role of capillary flow should be in a comprehensive theory. Line 47 with the phrase "getting away from" hints at an abandonment of capillary flow but that would contradict lines 61-62 which note that it works well under steady-state conditions. To more directly acknowledge the actual value and appropriate use of the capillary flow concept could make the commentary more appealing to readers whose thinking is deeply immersed in capillary concepts.

For the approach advocated here, the term "viscous flow" is a poor choice, though I realize there is precedent in the literature for this usage. Viscous forces are at play in every type of fluid flow through a medium, and thus viscosity is fundamental to liquid flow no matter what the driving force is. Viscosity appears explicitly in Richards' (1931) equations 3-6, and implicitly, as a fundamental element of hydraulic conductivity, in equations that follow. Capillary force, like gravitational force, does not turn off viscous forces but acts in combination with them. Whether driven by capillarity alone, gravity alone, or a combination of both, unsaturated flow is always influenced by viscosity. A better choice would be to distinguish this type of flow with a term like gravity-driven, or noncapillary. I don't see a perfect choice for this, but it should be a term that is not so broad that it literally includes flow types (Richards' for example) that are meant to be excluded.

Additional points by line number:

17-19. To start with the observation concerning the environment of the root tips of plants as requiring the effects of capillarity and unsaturation is an interesting way to stimulate reader interest.

55-66. The description in lines 56-59 of the processes of rainfall incidence is valuable. That these processes occur at positive or zero pressure is important for understanding the initiation of infiltration, runoff, and preferential flow. Germann's further explanation (lines 60-66), however, is confusing, perhaps largely because of the switch from the vivid concept of drops to abstractions like strong gradients and weak resistance, which are harder to pin on specific components of the soil-air-water system. If the two papers cited here are retained, they need some further explanation to give readers an idea of how acoustic velocity can relate to film thickness. I recommend either explaining more completely or omitting this paragraph.

60, 77, 79. Concerning the strength of driving forces, the labeling of one as the strong force and the other as the weak force can be confusing. In terms of fundamental forces, capillarity falls in the electrostatic category, which is stronger than the gravitational category. But in unsaturated flow, capillary force depends on the gradient of matric potential, and thus is sometimes greater than the gravitational force and sometimes weaker. The key distinction is that we have a frequently occurring mode of flow in which gravity is dominant and capillarity is negligible, yet common practice is to irrelevantly employ capillarity-based concepts in formulating it.

72-74. Instead of a blanket reference to all of the author's previous publications, readers would be better served by mention of a few of the most important ones, with a sentence or so noting the main import of each.

77-79. The statement citing Beven's (2018) paper is confusing because it is not clear what is being bridged to what, as well as because it implies that viscous flow is a type of force.

————-

Beven, K.: A Century of Denial: Preferential and Nonequilibrium Water Flow in Soils, 1864-1984, Vadose Zone Journal, 17, 10.2136/vzj2018.08.0153, 2018.

[Figure]

Briggs, L. J.: The mechanics of soil moisture, U.S. Department of Agriculture, Washington, DC, Bulletin 10, 24, 1897.

Buckingham, E.: Studies on the movement of soil moisture, U.S. Department of Agriculture, Washington, DC, Bulletin 38, 61, 1907.

King, F. H.: Observations and Experiments on the Fluctuations in the Level and Rate of Movement of Ground-water on the Wisconsin Agricultural Experiment Station Farm and at Whitewater, Wisconsin, U.S. Department of Agriculture Washington, DC, Bulletin 5, 75, 1892.

Richards, L. A.: Capillary conduction of liquids through porous materials, Physics, 1, 318-333, 1931.

Richardson, L. F.: Weather Prediction by Numerical Process, Cambridge University Press, Cambridge, UK, 262 pp., 1922.